# Comparative Analysis of Risk Factors in Declined Kidneys from Donation after Brain Death and Circulatory Death

**DOI:** 10.3390/medicina56060317

**Published:** 2020-06-26

**Authors:** Zinah Zwaini, Meeta Patel, Cordula Stover, John Dormer, Michael L. Nicholson, Sarah A. Hosgood, Bin Yang

**Affiliations:** 1Department of Infection, Immunity and Inflammation, University of Leicester, University Road, Leicester LE1 7RH, UK; zinahzwain@gmail.com (Z.Z.); Cms13@le.ac.uk (C.S.); 2Department of Infection, Immunity and Inflammation, Transplant Group, University of Leicester, Leicester General Hospital, Leicester LE5 4PW, UK; mp598@leicester.ac.uk; 3School of Medicine, University of Leicester, University Road, Leicester LE1 7RH, UK; john.dormer@leicester.ac.uk; 4Department of Surgery, University of Cambridge, Addenbrooke’s Hospital, Cambridge CB2 0QQ, UK; mln31@cam.ac.uk (M.L.N.); sh744@cam.ac.uk (S.A.H.); 5Department of Cardiovascular Sciences, University of Leicester, University Hospitals of Leicester, Leicester LE1 7RH, UK

**Keywords:** complement activation, cold ischemic time, donation after brain death, donation after circulatory death, inflammation, tissue injury, warm ischemic time

## Abstract

*Background and objectives:* Kidneys from donation after circulatory death (DCD) are more likely to be declined for transplantation compared with kidneys from donation after brain death (DBD). The aim of this study was to evaluate characteristics in the biopsies of human DCD and DBD kidneys that were declined for transplantation in order to rescue more DCD kidneys. *Materials and Methods:* Sixty kidney donors (DCD = 36, DBD = 24) were recruited into the study and assessed using donor demographics. Kidney biopsies taken post cold storage were also evaluated for histological damage, inflammation (myeloperoxidase, MPO), von Willebrand factor (vWF) expression, complement 4d (C4d) deposition and complement 3 (C3) activation using H&E and immunohistochemistry staining, and Western blotting. *Results:* More DBD donors (16/24) had a history of hypertension compared with DCDs (8/36, *p* = 0.001). The mean warm ischemic time in the DCD kidneys was 12.9 ± 3.9 min. The mean cold ischemic time was not significantly different between the two groups of kidney donors (DBD 33.3 ± 16.7 vs. DCD 28.6 ± 14.1 h, *p* > 0.05). The score of histological damage and MPO, as well as the reactivity of vWF, C4d and C3, varied between kidneys, but there was no significant difference between the two donor types (*p* > 0.05). However, vWF reactivity might be an early indicator for loss of tissue integrity, while C4d deposition and activated C3 might be better predictors for histological damage. *Conclusions:* Similar characteristics of DCD were shown in comparison with DBD kidneys. Importantly, the additional warm ischemic time in DCD appeared to have no further detectable adverse effects on tissue injury, inflammation and complement activation. vWF, C4d and C3 might be potential biomarkers facilitating the evaluation of donor kidneys.

## 1. Introduction

In the UK, approximately 12 to 15% of kidneys from deceased donors are retrieved, but then declined for transplantation [1]. The reasons for declined kidneys are mainly based on donor factors. Particular reservations are often noted with regard to the donation of kidneys after circulatory death (DCD), especially when other co-morbidities such as hypertension, diabetes, obesity and older age are present [2,3]. There is considerable discrepancy amongst different centres in the criteria for accepting a kidney donor, as powerful indicators for assessing the quality of donor kidneys and facilitating clinical decision-making are lacking [3]. This means that many kidneys might be unnecessarily discarded. Approximately 17% of DCD kidneys are rejected for transplantation compared with 6% of kidneys from donation from brain death (DBD) donors in 2012 [1].

One of the major risk factors in DCD donor kidneys is the impact of warm ischemic injury. DCD kidneys have higher rates of delayed graft function (DGF) and a higher risk of primary non-function (PNF) compared with DBD kidneys [4,5,6,7]. The warm ischemic insult makes them more susceptible to cold ischemic injury that heightens the severity of ischemia-reperfusion injury (IRI). This causes early graft dysfunction and increases the likelihood of acute rejection [8]. However, increasing evidence suggests that this does not appear to have an impact on graft survival [8,9,10].

There are also other factors in injury and inflammation that may affect long-term graft survival. Myeloperoxidase (MPO), a marker mainly for neutrophil granulocytes, catalyses the formation of reactive nitrogen species and is elevated in patients with chronic kidney disease and associated with its progression [11]. Hypothermic machine perfusion reduced the inflammatory reaction by downregulating the expression of MPO in a reperfusion model of DCD [12]. The depletion of leucocytes in the hemoperfusate reduced the infiltration of MPO-positive cells and improved the renal function of porcine kidneys [13].

The von Willebrand factor (vWF) binds to the surface of endothelium when disrupting the endothelium-exposed sub-endothelial matrix. Under arterial shear conditions, platelets are tethered onto the immobilized vWF along the surface of the injured vessel wall, leading to platelet adhesion and thrombus formation at the site of injury. Platelet–vWF interactions are markedly abnormal in patients with end-stage renal disease, which was normalised by a successful kidney transplantation [14].

The complement system is an important mediator of IRI and allograft rejection [15]. Renal proximal tubular epithelial cells are especially sensitive to IR-induced injury [16,17]. Renal IRI and subsequent acute tubular necrosis can activate the complement system causing further injury [15,18]. Complement 3 (C3) is in the centre of the complement activation pathway and is produced locally within the kidney [15]. In addition, the thioester bonded C4 activation products C4d and epithelial C4d reactivity have been described in allograft rejection and IgA nephropathy [19,20].

The aim of this study was to reduce unnecessarily discarding DCD kidneys by systematically evaluating donor characteristics and assessing the level of injury, inflammation and complement activation in biopsy samples from DBD and DCD kidneys that were retrieved but then declined for transplantation. Certain injury parameters such as vWF reaction, C4d deposition and C3 activation might be potential biomarkers to be used as the judging criteria of donor kidney quality and facilitate clinical decision-making.

## 2. Materials and Methods

### 2.1. Ethics

From October 2012 to October 2013, 64 kidneys were deemed unsuitable for transplantation by the National Organ Allocation Scheme and were then recruited to this laboratory research project. The consent from the donor family for the use of the organs for transplant and research was obtained by the transplant coordinators before organ retrieval. Ethical approval was granted for the study by the National Research Ethics Commission in the UK (REC: 12/EM/0143, UHL 77018, Principle Investigator: Professor Michael L Nicholson). The definition of expanded criteria donor (ECD) included any donor aged ≥ 60 years or a donor aged ≥ 50 years plus two out of three of the following features: a history of hypertension, a raised terminal serum creatinine (>132 µmol/L) or death from a cerebrovascular accident.

### 2.2. Histological Evaluation

A single wedge biopsy was taken on arrival at the laboratory after a period of static cold storage. The tissue was divided and either fixed in 10% formal saline then embedded in paraffin wax, or frozen in liquid nitrogen and stored at −80 °C. Four µm paraffin sections were stained with haematoxylin and eosin (H&E) for histopathological scoring or anti MPO, vWF and C4d antibodies for immunohistochemical analysis.

H&E-stained sections were assessed using the Remuzzi score by a consultant pathologist who was blinded to the donor types. Four different parameters, glomerular global sclerosis, tubular atrophy, interstitial fibrosis and vascular lesions, were used in the scoring system [21]. The score ranged from a minimum of 0 (indicating the absence of renal lesions) to 3 (severe). The sum of the four parameters was then calculated, with a score of 0–3, 4–6 and 7–12, indicating mild, moderate and severe changes, respectively. Sections were also graded mild, moderate and severe by the presence of acute tubular injury.

### 2.3. Myeloperoxidase Activity

Immunohistochemical staining of myeloperoxidase activity (MPO), a marker mainly for neutrophil granulocytes, was undertaken on paraffin sections using a DAKO ChemMate EnVisionTM Detection Kit (DAKO, Glostrup, Denmark). The sections were digested by 40 µg/mL proteinase K for 15 min at 37 °C, then blocked by peroxidase-blocking reagent. The sections were labelled by an anti-MPO antibody (1:600, DAKO) at 4 °C overnight. The antibody binding was revealed by a secondary antibody and 3′-amino-9-ethylcarbazole (AEC, dark red colour). MPO + cells in tubular, interstitial and glomerular areas were semi-quantitatively scored by counting the number of positive cells in 20 fields at 400× magnification.

### 2.4. Glomerular Accumulation of vWF Reactivity

vWF reactivity detected by immunohistochemical staining in the glomerular areas of the declined kidneys was assessed under light microscope at 100× and 400× magnification and classified according to its intensity (nil, mild, moderate and strong). Normal glomeruli were not stained by vWF [22]. Two independent observers blinded to the sample groups assessed the sections using predefined criteria. Image Pro Plus and Q imaging microPublisher 5.0 RTV installed in the Olympus CX41 light microscope were used to capture pictures and evaluate the staining intensity.

### 2.5. Epithelial Deposition of C4d

Immunohistochemical analysis was performed using a monoclonal antihuman C4d antibody as the primary antibody. The staining of C4d was examined under light microscope at 100× and 400× magnification. Two to three independent observers blinded to the sample groups evaluated the staining pattern and density (nil, minimal, focal and diffuse). The extent of tubular involvement for C4d deposition was qualitatively scored and was reproducible between observers.

### 2.6. Complement 3 in Kidney Lysates

Tissue samples obtained from kidney donors were processed for Western blotting analysis using a polyclonal anti C3 antibody, which reacts with native and activated C3. A total of 58 samples were analysed, so conditions (optimised antibody concentrations, amount of protein and exposure times) were initially standardised. The membranes were probed with β-actin as a loading control to normalise C3 reactivity.

### 2.7. Statistics

Continuous data were presented as mean ± SD, and range, where appropriate. Data were compared using the student’s *t*-test for parametric or Mann–Whitney U test for nonparametric variables. Categorical variables were analysed by Fisher’s exact test. A value of *p* < 0.05 was considered statistically significant. GraphPad Prism 6 was used for the statistical analysis (GraphPad Software, La Jolla, CA, USA).

## 3. Results

### 3.1. Description of Donated Kidneys

Sixty-four kidneys were recruited into this research study. Four kidneys were excluded: one for damage to the renal vessels and three for not being chilled in transit. Therefore, 60 kidneys (24 DBD and 36 DCD kidneys) were analysed. The donor demographics are listed in Table 1. The DBD and DCD groups included seven and eight pairs of kidneys, respectively. Seventy-five percent of the kidneys in the DBD group were from ECD compared with 56% in the DCD group (*p* = 0.174). There was no significant difference in the donor age (*p* = 0.878). In the DBD kidneys, death was caused by an intracranial haemorrhage (ICH) in 88% of donors compared with 33% in the DCD group (*p* < 0.0001). There were significantly more DBD donors (67%) who had a history of hypertension compared with the DCD group (22%, *p* = 0.001). The period of ventilation was significantly longer in the DCD donors (*p* = 0.008).

There was no significant difference in the serum creatinine (SCr, Table 1) level at the time of retrieval between the groups (*p* > 0.05). At retrieval, the level of SCr was increased >132 µmol/L in 6 DBD and 11 DCD donors (*p* = 0.575). The cold ischemic time (CIT, Table 1) was longer in the DBD kidneys (*p* = 0.263), which exceeded 30 h in 14 of the DBD and 14 of the DCD kidneys (*p* = 0.296). The mean warm ischemic time (WIT) in the DCD kidneys was 12.8 ± 3.9 min.

Kidneys were declined for transplantation for a variety of reasons (Table 2). The most common cause of decline in the DBD group was past medical history. This included a pair of kidneys from an older donor with haematuria, another pair from a donor with a history of hypertension, one kidney from a donor with a history of renal stones and five kidneys from donors with a suspected malignancy. In the DCD kidneys, past medical history and poor in situ flush were the commonest causes of decline. Past medical history included suspected malignancies in nine cases and one kidney from a donor with a raised SCr to 542 µmol/L at retrieval.

### 3.2. Histology and Inflammation

Six biopsies that had fewer than 25 glomeruli were excluded from the analysis. There was no significant difference in the Remuzzi scores between the DBD and DCD kidneys (*p* > 0.05; Table 3). Two kidneys in the series that were declined due to the histological evaluation, one in each of the DBD and DCD groups scored as moderate and severe, respectively, according to the Remuzzi score [6,7].

There was a strong positive correlation between WIT and histological damage in the DCD kidneys (*r* = 0.900, Figure 1). However, there was no significant relationship between CIT and histological score either in the DBD or DCD kidneys.

All kidneys were also graded for acute tubular injury (ATI), and there was no statistical difference between the DBD and DCD kidneys (*p* > 0.05). Seven DBD kidneys were scored mild and another three were moderate ATI, while the DCD kidneys were scored 9 mild, 10 moderate and 1 severe ATI. In addition, the mean number of MPO positive cells was similar between groups (DBD 27.2 ± 24.3 vs. DCD 37.0 ± 23.5/mm^2^, *p* = 0.197; graphic data not shown).

### 3.3. Glomerular Accumulation of vWF Reactivity

The reactivity of vWF was varied between different biopsies and was also uniform within the section of individual biopsy. Very strong (+++), moderate (++), mild (+) and negative staining of vWF were found in DBD kidneys and DCD kidneys, without significant differences between the two groups (all *p* > 0.05, Table 4).

The representative images of vWF staining at 100× (Figure 2a,b) and 400× (Figure 2c, glomerular capillary convolutes) magnifications are shown. In the DBD kidneys, the level of mild vWF reactivity was positively associated with histological score (*r* = 0.945), with which the level of moderate vWF was reversely related (*r* = 0.967, Figure 2d). However, this was not the case in the DCD kidneys. In addition, the percentage of moderate to strong vWF reaction was increased by prolonged CIT in both the DBD and DCD kidneys, and prolonged WIT in the DCD kidneys (Figure 2e).

### 3.4. Epithelial Deposition of C4d, A Marker of Complement Activation

There was no discrimination in C4d reactivity between the groups of DBD and DCD, with C4d deposition present around tubular epithelial cells (Figure 3a–d). Very similar diffuse deposition was found in these two groups, 21.7% in DBD vs. 21.2% in DCD, while a few kidney samples were negative for C4d, 8.7% in DBD vs. 9.1% in DCD.

In both the DBD and DCD kidneys, the similarly increased diffuse C4d deposition in the tubular areas was shown in line with increased histological score (Figure 3e,f). The percentage of C4d deposition was unexpectedly more in both DBD and DCD kidneys subjected to CIT ≤ 24 h, or DCD kidney with WIT < 16 min, than those with prolonged CIT or WIT (Figure 3g,h).

### 3.5. Complement C3 and β-Actin in Kidney Lysates

Two major bands, 115 and 68 kDa, representing the α and β chain of C3, respectively, were present in the kidney lysates samples (Figure 4a). The C3 expression was visualised in all DBD kidneys (*n* = 24) and 88.6% of DCD kidneys with four negative for C3 (*n* = 34). The reactivity with the polyclonal anti C3 antibody of bands less than 68 kDa, representing 43 kDa iC3b, 41 kDa C3dg and 27 kDa C3c, were recorded as C3 active products (Figure 4a). A total of 70.8% of the DBD samples were positive for C3 active products, while only 44.1% of the DCD samples were positive. The expression of C3 was more abundant in the DBD than that in the DCD kidneys, but there was no significant difference between the two donor types (Figure 4b).

Although β-actin is a well-known housekeeping protein, it was variable in this study (Figure 4a). It was also discordant with the Coomassie stain of a gel run in parallel, so could not be used for normalisation. The expression of β-actin was more abundant in the DCD kidneys than DBD, with a significant difference between two, while negative β-actin in the DCD kidneys was half of the DBD kidneys (Figure 4c).

Both DBD and DCD kidneys that scored as severe in the histopathological assessment were positive for C3. The abundant expression of C3 fragment was positively related to the severity of histological damage in the DBD kidneys (*r* = 0.969, Figure 4d). However, this relationship was not seen in the DCD kidneys for the C3 expression (Figure 4e), or both donor types for β-actin expression even though all DBD kidneys that scored severe in the histopathology were positive for β-actin.

The expression of C3 was stronger in both the DBD and DCD kidneys subjected to CIT ≤ 24 h (Figure 4f). The abundant expression of C3 was found in all DBD with CIT ≤ 24 h and 70% DBD with CIT > 24 h, and a similar change trend of C3 was seen in the DCD kidneys. In addition, the C3 expression was slightly increased by prolonged WIT (Figure 4g).

When relating the presence of C3 activation products and C4d deposition in kidney tissues, two parameters of complement activation, a greater proportion of DBD kidneys (69.6%) showed evidence of complement activation compared with the DCD kidneys (46.9%, Table 5). Interestingly, 9.4% of the DCD kidneys compared with 4.3% of the DBD kidneys showed neither C3 active products nor C4d deposition.

## 4. Discussion

Although DCD kidneys are more likely to be rejected for transplantation, this study showed similar levels of histologic damage, inflammation, cellular injury and complement activation in DBD and DCD donor kidneys that were declined for transplantation. The significant differences in clinical risk factors such as higher percentage of hypertension and intracranial haemorrhage in DBD donors, but additional warm ischemic time and prolonged ventilation in DCD donors, may be complementarily balanced in the outcome of detected injury parameters. In addition, vWF reactivity as an endothelial cell activation marker and complement activation might mirror the ischemic insult up to a certain degree and then decline when damage becomes severe, while C4d deposition and activated C3 might be predictors for concomitant histological damage.

In the UK, the number of DCD kidney transplants has risen over the years [1]. However, the proportion of kidneys that were retrieved but subsequently discarded has increased to 17% from 2003 to 2012 [1], which was in contrast to only 6% of DBD kidneys. Schemes such as the Fast Tracking System have been employed in the UK to efficiently utilise marginal kidneys, ensuring that they were rapidly allocated to minimise the cold ischemic time [2]. Nonetheless, there is still a great deal of inconsistency between transplant units in accepting kidneys and many are still being discarded unnecessarily.

Kidneys recruited in this study were discarded for a variety of reasons. Past medical history was the commonest cause of decline. Fourteen of these were from donors with suspected malignancies, which would not be considered for transplantation, but were included in the analysis. Inadequate in situ perfusion was also a common cause of decline particularly in the DCD kidneys (40%). This could result from improper placement of the aortic cannula, renal vasospasm or intravascular thrombosis [23,24]. A large proportion of both DCD and DBD kidneys in this series were from ECD donors. Donor age is one of the main risk factors for reduced graft survival and there has been some reluctance to use older DCD kidneys [25,26,27]. However, a recent report by Summers et al. showed that kidneys from DCD older donors have equivalent graft survival compared to older DBD donors [10].

We used a range of histological parameters to assess injury, inflammation and complement activation. The Remuzzi score was used to assess and grade the histological damage of the samples. This scoring system was originally developed to assess whether kidneys from marginal donors had enough viable nephrons to be transplanted [21]. Their recommendations were that kidneys with a score of 0–3 are suitable for a single transplant, those with a score of 4–6 suitable for a dual transplant and those with a score above 7 unsuitable for transplantation [21]. Eight kidneys had a score of 7 or above (four DBD and four DCD), indicating severe injury (Table 3), and would have been discarded on this basis. Interestingly, excluding the kidneys declined due to suspected malignancies, 15/55 (27%) of these kidneys scored 0–3 and would have been considered suitable for transplantation on this basis. Both DBD and DCD kidneys showed a similar level of acute tubular injury.

The expression of vWF in renal glomeruli has been reported in patients with hypertension or vascular disease [22]. A significant proportion of DBD donors in this series had hypertension compared with the DCD group. However, a mild to moderate expression of vWF in this study was equally found in both the DBD and DCD kidneys. In addition, the DCD kidneys did not show any increase in the level of neutrophil infiltration measured by myeloperoxidase activity.

Cold ischemia is another major risk factor for DGF and reduced graft survival [10,28,29,30]. The cumulative effect of warm and cold ischemic injury is particularly detrimental to graft survival [6]. In this series, the cold ischemic time was prolonged in many of the kidneys. Cold ischemia also influences complement activation and increases the likelihood of rejection. De Vries et al. found that after the transplantation of DCD and DBD kidneys, there was acute evidence of complement activation (sC5b-9), possibly originating from cold ischemic damage to the endothelium [31]. C4d deposition has also been found in tubular epithelial cells of deceased donor kidneys. Naesens et al. found a significant correlation between the cold ischemic time and expression of complement genes [32]. However, the level of C3 synthesis only reached a peak after reperfusion was established in the transplanted organ, within 48 h of surgery [15,18]. The renal tubular epithelial cells sustain the most damage, owing to intense production of C3 in the hypoxia-sensitive region of the kidney [32]. In this study, the majority of kidneys showed evidence of endothelial and epithelial damage and complement activation. However, importantly, we did not reveal a substantial difference between the donor types.

Brain death is viewed by some as a risk factor for poor allograft survival causing variations in the catecholamine level that could lead to hypoxia and inflammation in the donor organ. In this series, the majority of the DBD donors suffered intracranial haemorrhage (88%) and hypertension (67%) compared with 33% and 22% of the DCD donors, respectively. Conversely, DCD donors were subjected to significantly prolonged ventilation (3.6 times) compared with DBD donors. Damman et al. found an increased C3 mRNA expression in kidney biopsies from DBD compared with DCD donors [33]. However, we found a similar level of complement and inflammation in the DCD and DBD kidneys.

This study aimed to distinguish characteristics of biopsy samples taken from DBD and DCD donor kidneys that had been declined for transplantation. Clinically, the distinct difference between DBD and DCD kidneys is the higher rate of DGF in DCD kidneys, yet this argument does not affect graft survival. The pathogenesis of DGF in DCD kidneys was triggered by the additional warm ischemic insult. However, our findings suggest that the DCD kidneys declined for transplantation showed a similar level of histological injury (Table 3), inflammation (MPO + cells), complement C4d deposition and C3 activation compared to DBD kidneys.

The outcome from the discarded donor kidneys in this study is in agreement with a report by Damman et al. that showed similar hypoxic, complement and coagulation pathways in DBD and DCD donors [34]. Although this study was limited to assessing the biopsy characteristics of DCD kidneys in comparison with DBD kidneys after a period of static cold storage, our group showed that ex vivo normothermic machine perfusion (NMP) enabled an assessment and restoration of the DCD kidneys that had been declined for transplantation due to inadequate in situ perfusion [35]. Furthermore, our group reported that the use of ex vivo NMP after normothermic regional perfusion improved a poorly perfused DCD kidney that was then transplanted into a suitable recipient. The recipient had slow early graft function, but did not require dialysis posttransplant, and was discharged six days later with an SCr of 160 μmol/L (1.8 mg/dL) at two months posttransplant [36]. Therefore, discarded DCD donor kidneys, at least some of them, could be salvaged by NMP. The supportive evidence also demonstrated that DCD kidneys from donors dying of ligature asphyxiation suffered an additional warm ischemic insult that does not adversely influence transplant outcomes [37].

## 5. Conclusions

This study has shown a great deal of variability in the quality of DBD and DCD kidneys declined for transplantation. There were no significant differences in the histologic damage, cellular injury, inflammation and complement activation renal biopsies of the DBD and DCD kidneys. vWF reactivity and complement activation might be early indicators for a certain degree insult, while C4d deposition and C3 activation might be also good predictors for histological damage. These potential biomarkers in conjunction with NMP could be applied to better assess and rescue viable DCD donor kidneys.

## Figures and Tables

**Figure 1 medicina-56-00317-f001:**
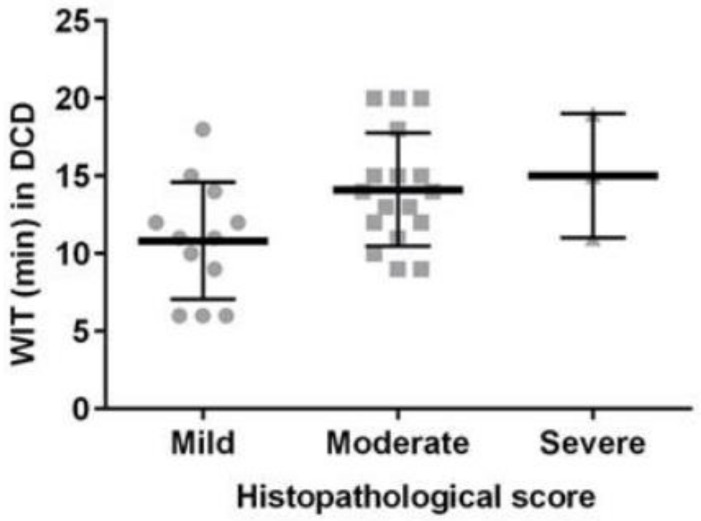
The relationship between warm ischemic time (WIT) and histological damage in donation after brain death (DBD) and donation after circulatory death (DCD) kidneys.

**Figure 2 medicina-56-00317-f002:**
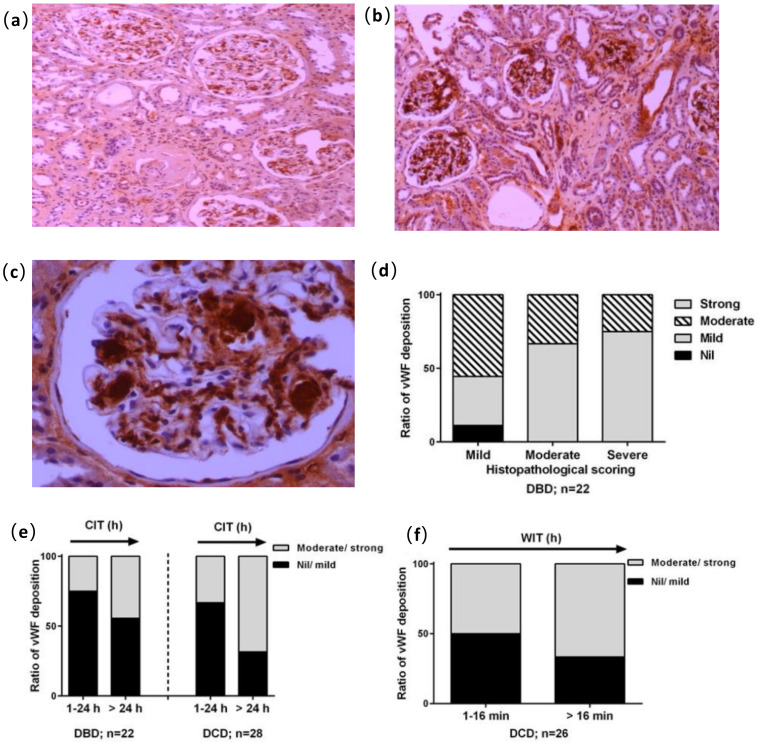
Immunostaining of von Willebrand factor (vWF) and reactivity analysis in glomerular areas. Examples of subjective moderate and strong staining in reaction intensity ((**a**,**b**), 100×) and strong vWF reactivity appeared within capillary convolutes ((**c**), 400×). (**d**) vWF reactivity plotted against histological changes in DBD and DCD kidneys. (**e**) The effect of cold ischemic time (CIT) within 24 h or more than 24 h on vWF deposition in the DBD and DCD kidneys. (**f**) Semi quantitative evaluation of the DCD kidneys for vWF staining plotted against WIT.

**Figure 3 medicina-56-00317-f003:**
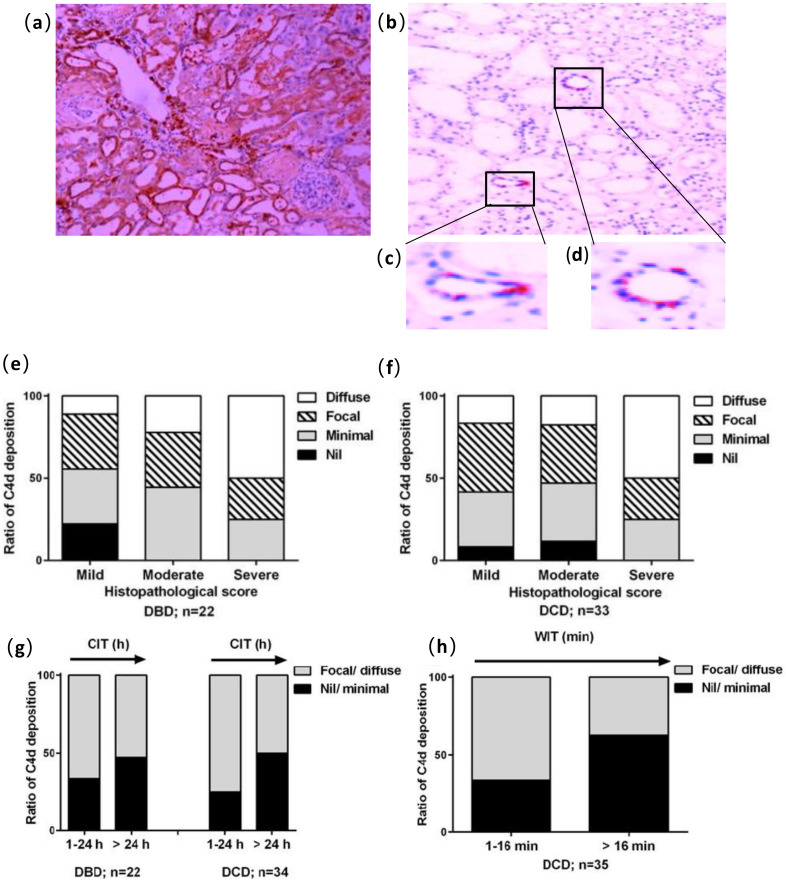
Immunostaining analysis of C4d deposition in tubular epithelia (100×). (**a**) Diffuse deposition; (**b**) focal deposition; (**c**,**d**) enlarged pictures demonstrating the deposition of C4d around tubular epithelial cells; (**e**,**f**) C4d reactivity plotted against histological changes in DBD and DCD kidneys (*n* = 23 and 27, respectively); (**g**) The effect of CIT ≤ 24 h or > 24 h on C4d deposition in DBD and DCD kidneys; (**h**) WIT-affected C4d deposition in DCD kidneys.

**Figure 4 medicina-56-00317-f004:**
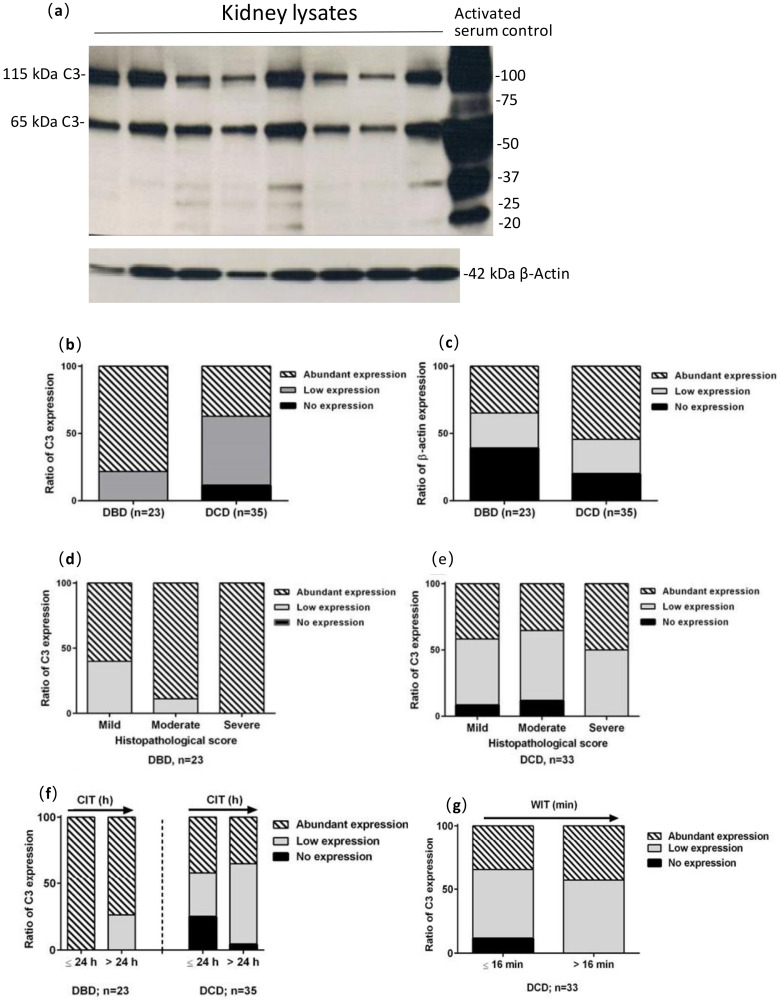
Semi-quantitative analysis of complement C3 and β-actin in DBD and DCD kidneys. (**a**) An exemplary autoradiograph of Western blotting analysis of human kidney lysates showing two major bands for complement C3 (C3, 115 and 65 kDa) and lower weight C3 products in some samples (activated serum loaded for comparison). β-actin (42 kDa) reactivity was also revealed (lower part of figure); (**b**,**c**) semi quantitative analysis of C3 and β-actin in DBD and DCD kidneys; (**d**,**e**) C3 expression was plotted against histological score in DBD and DCD kidneys; (**f**,**g**) the expression of C3 in DBD and DCD kidneys plotted against CIT and WIT.

**Table 1 medicina-56-00317-t001:** Donor demographics such as cause of death and hypertension.

	DBD (*n* = 24)	DCD (*n* = 36)	*p* Value
Age (y)	61 ± 15	61 ± 12	0.878
Male: Female	10:14	24:12	0.068
ECD	18 (75%)	20 (56%)	0.174
Ethnic group			
White	22 (92%)	35 (97%)	0.558
Other	2 (8%)	1 (3%)	
Cause of Death			
ICH	21 (88%)	12 (33%)	<0.0001
Hypoxia	3 (13%)	11 (30.6%)	0.13
Other	0	13 (37.1%)	0.001
BMI	28.2 ± 4.6 (21.8–33.9)	27.8 ± 7.4 (18.4-54.8)	0.491
Hypertension	16 (67%)	8 (22%)	0.001
Smoker	12 (50%)	12 (33.3%)	0.283
Diabetes	2 (8%)	0	0.156
Episode of cardiac arrest	10 (42%)	20 (56%)	0.430
Cardiac arrest (min)	29.9 ± 18.2 (3–60)	24.2 ± 13.8 (9–45)	0.218
Ventilation (min)	2413 ± 1692	8598 ± 14722	0.008
Inotropes	20 (83%)	24 (67%)	0.234
Hypotensive period	19 (79%)	24 (67%)	0.385
Hypertensive period	13 (54%)	14 (39%)	0.295
Retrieval Cr (µmol/L)	103 ± 61 (61–271)	136 ± 102 (49–542)	0.182
Urine output last h (ml)	78 ± 59	90 ± 100	0.758
CIT (h)	33.3 ± 16.7 (7.4–76.5)	28.6 ± 14.1 (6.3–71.6)	0.263
Agonal phase (min)	-	36.1 ± 42.2 (0–152)	
WIT (min)	-	12.8 ± 3.9 (6–20)	

ICH: intracranial haemorrhage, DBD: donation after brain death, DCD: donation after circulatory death, BMI: body mass index, ECD: extended criteria donation, SCr: serum creatinine, CIT: cold ischemic time, WIT: warm ischemic time.

**Table 2 medicina-56-00317-t002:** Reasons for declined kidney donors such as poor flush and histology score.

Reasons for Decline	DBD (*n* = 24)	DCD *(n* = 36)	*p* Value
PMH	9 (38%)	11 (31%)	0.418
Poor flush	3 (13%)	10 (28%)	0.210
Donor age	6 (25%)	3 (8%)	0.137
HMP parameters	1 (4%)	4 (11%)	0.639
Technical/anatomical	2 (8%)	4 (11%)	1.000
Histology score	1 (4%)	1 (3%)	1.000
Prolonged CI	1 (4%)	3 (8%)	0.634
No suitable recipient	1 (4%)	0 (0%)	0.400

PMH: past medical history, HMP: hypothermic machine perfusion and CI: cold ischemia.

**Table 3 medicina-56-00317-t003:** Histological changes scored by system.

Remuzzi Score	DBD *(n* = 23)	DCD (*n* = 31)	*p* Value
<3 (mild)	11 (47.8%)	10 (31.3%)	1.000
4–6 (moderate)	8 (34.8%)	17 (53.1%)	0.272
>7 (severe)	4 (17.4%)	4 (12.5%)	0.707

Four parameters (glomerular sclerosis, tubular atrophy, interstitial fibrosis and vascular lesions) were assessed in H&E-stained sections, with the score range from 0 to 3 summed up for each parameter.

**Table 4 medicina-56-00317-t004:** vVW staining in DBD kidneys and DCD kidneys.

	DBD (*n* = 23)	DCD (*n* = 27)	*p* VALUE
vWF+++	0 (0%)	4 (14.8%)	>0.05
vWF++	10 (43.5%)	11 (40.7%)	>0.05
vWF+	12 (52.2%)	9 (33.3%)	>0.05
vWF−	1 (4.3%)	3 (11%)	>0.05

vWF: von Willebrand factor.

**Table 5 medicina-56-00317-t005:** Activated C3 and/or C4d deposition in DBD and DCD kidneys.

	DBD (*n* = 23)		DCD (*n* = 32)	
C4d+	21 (91.3%)		29 (90.6%)	
C3+	16 (69.6%)		15 (46.9%)	
C3−		5 (21.7%)		14 (43.7%)
C4d−	2 (8.7%)		3 (9.4%)	
C3+	1 (4.3%)		0 (0%)	
C3−		1 (4.3%)		3 (9.4%)

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
