# Peer review of "Comparative Analysis of Risk Factors in Declined Kidneys from Donation after Brain Death and Circulatory Death"

_medicina, 2020, doi:10.3390/medicina56060317_

Round 1

Reviewer 1 Report

The papers methodology is of interest and the authors went to great lengths to analyse the two kidney donor groups for histopathological damage and markers of inflammation and compliment activation. They have outlined in the introduction the aim of the study was to contrast between DBD and DCD donors. In the results however they have failed on numerous occasions to do this, making the results confusing and difficult to understand. It is critical that if there was no statistically significant difference found between the two groups that this be all that is said, rather than suggesting a numerical difference or trend is of interest.  Much of the data could be presented in a table form rather than the numerous small graphs,  particularly for those variables that were shown to be not of statistical significance. The pictures of the histopathology are unhelpful and not necessary. The conclusion seems to duplicate some aspects of the methodology and results rather than discuss the original hypothesis and the literature that is currently available.

The numerous small graphs are confusing and difficult to follow. I would focus on contrasting between the two donor groups rather than differences within the sub classes of donor groups. It does seem that the original results suggests no difference between the two classes of donors in the variables measured which is of some interest. Given the difficulty in accessing rejected kidneys the results are worth publishing once the results are simplified and better presented.  There are several typographical error errors which could be corrected with a more thorough proof read.

The  differences in the clinical parameters between two groups that was shown to be statistically significant are not really highlighted well in the discussion. Of interest is the significant difference between the length of ventilatory support provided to the DCD donors. These this will certainly have an impact on a variety of variables that should be further discussed. The WIT was very short in the DCD, with the maximum value of 20 minutes. This may have  minimised the amount of damage visualised in the DCD kidneys versus DBD kidneys and reduced the  impact of warm ischaemia time on the histological variables measured.

The paper is of interest in that it is an important area of study with unique access to a large cohort of rejected kidneys. It is certainly worthwhile to attempt to salvage this paper with fewer and more specific and focussed diagrams, a clear table of results and improved discussion, with less histopathology pictures and simplified results section particularly for the many variables that were not found to be statistically different between the two groups. A clear focus on the differences between the two groups rather than subclasses is needed to make the paper readable.

Author Response

To Reviewer 1:

The papers methodology is of interest and the authors went to great lengths to analyse the two kidney donor groups for histopathological damage and markers of inflammation and compliment activation. They have outlined in the introduction the aim of the study was to contrast between DBD and DCD donors. In the results however they have failed on numerous occasions to do this, making the results confusing and difficult to understand.

Lots of changes were made throughout the manuscript, such as delete some confusion data and added clear statements "without significant differences between two donor types".

It is critical that if there was no statistically significant difference found between the two groups that this be all that is said, rather than suggesting a numerical difference or trend is of interest.  Much of the data could be presented in a table form rather than the numerous small graphs,  particularly for those variables that were shown to be not of statistical significance. The pictures of the histopathology are unhelpful and not necessary.

More than half of histopathologic pictures in Figure 2 and 3 were deleted and only remained ones that we think are necessary. 

The conclusion seems to duplicate some aspects of the methodology and results rather than discuss the original hypothesis and the literature that is currently available.

The conclusion was changed.

The numerous small graphs are confusing and difficult to follow. I would focus on contrasting between the two donor groups rather than differences within the sub classes of donor groups. It does seem that the original results suggests no difference between the two classes of donors in the variables measured which is of some interest. Given the difficulty in accessing rejected kidneys the results are worth publishing once the results are simplified and better presented. There are several typographical error errors which could be corrected with a more thorough proof read.

Some small graphs were deleted such as original Figure 1a, Figure 2f, h, Figure 3h, Figure 4b,c and Figure 5b, d; and a couple of others Figure 2f and 5e were changed into Table 4 and 5 respectively.

Typographical errors, as well as other corrections, were made for easy to read.

The  differences in the clinical parameters between two groups that was shown to be statistically significant are not really highlighted well in the discussion. Of interest is the significant difference between the length of ventilatory support provided to the DCD donors. These this will certainly have an impact on a variety of variables that should be further discussed.

Significantly changed clinical parameters were added in the section of discussion, such as intracranial haemorrhage hypertension and the length of ventilator support.

The WIT was very short in the DCD, with the maximum value of 20 minutes. This may have  minimised the amount of damage visualised in the DCD kidneys versus DBD kidneys and reduced the  impact of warm ischaemia time on the histological variables measured.

The paper is of interest in that it is an important area of study with unique access to a large cohort of rejected kidneys. It is certainly worthwhile to attempt to salvage this paper with fewer and more specific and focussed diagrams, a clear table of results and improved discussion, with less histopathology pictures and simplified results section particularly for the many variables that were not found to be statistically different between the two groups. A clear focus on the differences between the two groups rather than subclasses is needed to make the paper readable.

We very much appreciated the positive and valuable comments from the reviewer 1 and believe that we improved diagram, result presentation and discussion etc. and hope this manuscript is now acceptable for publication by Medicina.

Reviewer 2 Report

Overall comments to the Author

Thank you for the opportunity to review the manuscript entitled, "Comparative analysis of risk factors in declined kidneys from donation after brain death and circulatory death". I think that the manuscript is well written. However, this study contains several drawbacks, especially in description of the aim of the study. I have the following comments for major revision:

Major

  1. The authors describe that the aim of this study was to evaluate donor characteristics and assess the histological findings in DBD and DCD kidneys. However, the meaning of this aim is unclear. The authors should mention what kinds of meanings the results of this study have, especially in clinical settings.
  2. I agree with the authors with regard to “some kidneys are discarded unnecessarily”. How do the present results contribute to the improvement of the unnecessary rejection for transplantation?

Author Response

To Reviewer 2:

Major:

1. The authors describe that the aim of this study was to evaluate donor characteristics and assess the histological findings in DBD and DCD kidneys. However, the meaning of this aim is unclear. The authors should mention what kinds of meanings the results of this study have, especially in clinical settings.

We appreciate this comment and added “in order to rescue more DCD kidneys” in the aim of this study.

2. I agree with the authors with regard to “some kidneys are discarded unnecessarily”. How do the present results contribute to the improvement of the unnecessary rejection for transplantation?

We appreciate this comment and added “vWF, C4d and C3 might be potential biomarkers facilitating the evaluation of donor kidneys” .

Round 2

Reviewer 1 Report

There remains a typo on line 178-should read history not histology. 

Author Response

We thanks the reviewer 1 accepted our previous revision and changed “histology” to “history”.

We also made a few additional typo changes as the following:

Line 13: deleted “,” after “School of Medicine”.

Line 32 & 72: Changed “von Willebrand Factor” to “von Willebrand factor”.

Line 61: Changed “warm iscemic insult”to “warm ischemic insult”.

Line 66: Changed “catalyses” to “catalysing”.

Line 342: Changed “repectively” to “respectively”.

Reviewer 2 Report

Thank you for the opportunity to review the manuscript again, but unfortunately, the introduction and discussion sections have not been fully revised. I understand it's very important to reduce unnecessary organ discard, but I'm afraid that the design and results of this study do not still contribute to the reduction of it.

Author Response

We fully agree with the persisting in the aims of this study from the reviewer 2, and do apology for the first revision in which only the abstract was changed in aims, but not introduction. In this revision, we rephrased the last paragraph of the introduction (Line 83-88), and also added “Certain injury parameters such as vWF reaction, C4d deposition and C3 activation might be potential biomarkers to be used as the judging criteria of donor kidney quality and facilitating clinical decision-making”. We hope with these changes better shaped the design of the study and clarified the aims for the study, as well as reflected the meaning of available results especially in clinical settings.

In the section of discussion (Line 288-291): Added “The significant differences in clinical risk factors such as higher percentage of hypertension and intracranial haemorrhage in DBD donors, but additional warm ischemic time and prolonged ventilation in DCD donors, may be complementarily balanced the outcome of detected injury parameters. In addition…”.

We also changed (Line 372-373): “These potential biomarkers could be applied to better assess and rescue viable DCD donor kidneys” to “These potential biomarkers in conjunction with NMP could be applied to better assess and rescue viable DCD donor kidneys”.

We made a few other changes as well:

Line 57-58 & 296: The percentage of discarded DBD and DCD kidney was change from “8%” to “6%”, and “18%” to “17%“ more specifically referred to 2012 according to reference [1].

Line 295-296: Changed “over a period of time” to “from 2003 to 2012”.

Line 357: Added “our group showed that”.

Line 357: Added “we reported that”.

Please advise if there were any further comments. 

Round 3

Reviewer 2 Report

The authors have fully revised the manuscript. I have no further comments on it.